# Superexchange-stabilized long-distance Cu sites in rock-salt-ordered double perovskite oxides for $CO_2$ electromethanation

Jiawei Zhu [⊕][1,2,3,10] [✉], Yu Zhang[1,2,3,4,10], Zitao Chen [⊕][5], Zhenbao Zhang[6], Xuezeng Tian [⊕][5], Minghua Huang [⊕][7], Xuedong Bai [⊕][5], Xue Wang [⊕][8], Yongfa Zhu [⊕][9] & Heqing Jiang[1,2,3] [✉]

Cu-oxide-based catalysts are promising for $CO_2$ electroreduction ($CO_2$RR) to $CH_4$, but suffer from inevitable reduction (to metallic Cu) and uncontrollable structural collapse. Here we report Cu-based rock-salt-ordered double perovskite oxides with superexchange-stabilized long-distance Cu sites for efficient and stable $CO_2$-to-$CH_4$ conversion. For the proof-of-concept catalyst of $Sr_2CuWO_6$, its corner-linked $CuO_6$ and $WO_6$ octahedral motifs alternate in all three crystallographic dimensions, creating sufficiently long Cu-Cu distances (at least 5.4 Å) and introducing marked superexchange interaction mainly manifested by O-anion-mediated electron transfer (from Cu to W sites). In $CO_2$RR, the $Sr_2CuWO_6$ exhibits significant improvements (up to 14.1 folds) in activity and selectivity for $CH_4$, together with well boosted stability, relative to a physical-mixture counterpart of $CuO/WO_3$. Moreover, the $Sr_2CuWO_6$ is the most effective Cu-based-perovskite catalyst for $CO_2$ methanation, achieving a remarkable selectivity of 73.1% at 400 mA cm$^{-2}$ for $CH_4$. Our experiments and theoretical calculations highlight the long Cu-Cu distances promoting *CO hydrogenation and the superexchange interaction stabilizing Cu sites as responsible for the superb performance.

$CO_2$ electroreduction ($CO_2$RR) into value-added chemical feedstocks and fuels, driven by local-generated renewable energy, is a highly promising strategy for realizing the carbon-neutral cycle together with earning potential economic returns[1–4]. Among all $CO_2$RR products, $CH_4$ is of considerable interest based on its well-established infrastructure toward storage, distribution, and utilization[5,6]. Up to date, mainly Cu-based catalysts are able to generate appreciable $CH_4$ via stabilizing and subsequently hydrogenating the *CO species during $CO_2$RR[7]. However, owing to the involvement of complicated 8-electron transfer steps and structural degradations (e.g., fragmentation,

dissolution, agglomeration), most Cu-based catalysts (e.g., oxide-derived Cu) still suffer from unsatisfactory Faradaic efficiency for $CH_4$ and poor stability[8,9].

Perovskite oxides (typically $ABO_3$), featuring distinct merits (e.g., diverse chemical compositions, flexible crystal and electronic structures, and governable physicochemical properties), have provided an attractive platform for accessing high-performance catalysts toward numerous electrochemical reactions[10–13]. Upon most occasions, the nature of B-site cations or B−O bonding determines the electrocatalytic properties of perovskite oxides in essence[14–16]. Based on the

[1]Qingdao Institute of Bioenergy and Bioprocess Technology, Chinese Academy of Sciences, 266101 Qingdao, China. [2]Shandong Energy Institute, 266101 Qingdao, China. [3]Qingdao New Energy Shandong Laboratory, 266101 Qingdao, China. [4]University of Chinese Academy of Sciences, 100049 Beijing, China. [5]Beijing National Laboratory for Condensed Matter Physics and Institute of Physics, Chinese Academy of Sciences, 100190 Beijing, China. [6]School of Chemistry and Chemical Engineering, Linyi University, 276005 Linyi, China. [7]School of Materials Science and Engineering, Ocean University of China, 266100 Qingdao, China. [8]School of Energy and Environment, City University of Hong Kong, 999077 Hong Kong, China. [9]Department of Chemistry, Tsinghua University, 100084 Beijing, China. [10]These authors contributed equally: Jiawei Zhu, Yu Zhang. [✉]e-mail: zhujw@qibebt.ac.cn; jianghq@qibebt.ac.cn

above characteristics of Cu-based catalysts and perovskite oxides, if the B sites could be occupied entirely or partly by Cu element, the corresponding Cu-based perovskite oxides would be active toward $CO_2$ electromethanation[17–22]. Typical examples involve Cu-based Ruddlesden–Popper perovskite oxides (e.g., $La_2CuO_{4−δ}$)[17–22]. Nonetheless, these catalysts with B sites wholly occupied by Cu exhibit low activity and selectivity for $CH_4$, owing to the distance of their adjacent Cu sites not far enough to inhibit the competitive C–C coupling[17–22]. Furthermore, like traditional Cu-based oxides (e.g., CuO and $Cu_2O$)[23–25], since the electrode-supplied electrons attack or break the Cu–O bond to reduce the Cu sites, these catalysts also undergo uncontrollable reconstructions (e.g., metallic Cu exsolution) during $CO_2RR$[20,21]. Such reconstructions could make the active sites unmaintainable, causing lowered catalytic performance or even deactivation[8,26,27].

Substitution of another cation (B') for B to form doped perovskite oxides (e.g., $AB_{1−x}B'_xO_3$) has been intensively proved as a tried-and-true strategy to optimize the catalytic performance of perovskite oxides[10,11,14–16]. Accordingly, for Cu-based perovskite oxides, partly occupying their initial Cu sites by the doping cations (B') could also availably modulate or enhance their catalytic properties toward $CO_2$-to-$CH_4$ conversion. In general, if the B' and Cu cations are almost equal in molar content, while they are sufficiently different in size and/or charge, Cu-based double perovskite oxides ($A_2CuB'O_6$) with B-site rock-salt ordering will be produced[28–30]. The formation of a double perovskite structure is very likely to introduce important benefits to the physicochemical properties, affecting activity, selectivity, and stability in $CH_4$ production[28–34]. Specifically, in the rock-salt-type arrangement, the B-site cations alternate in all three crystallographic dimensions, markedly widening the distance between adjacent Cu cations, theoretically almost doubling relative to the undoped ones[28–30]. This increased distance could suppress *CO dimerization and promote activity and/or selectivity for $CH_4$ production[31,32]. Moreover, the B-site rock-salt ordering could bring superexchange interaction between Cu and B' cations (mediated by intermediate O anions) and give rise to the redistribution of charge densities of the B-site cations via electron transfer[33,34]. During $CO_2RR$, this superexchange interaction may transfer the electrode-supplied electrons accumulated around the Cu sites to B' sites and stabilize the Cu sites, thereby boosting the catalytic stability. However, to our knowledge, such Cu-based double perovskite oxides have not been reported in $CO_2RR$, so the vital roles of their unique physicochemical properties in catalytic performance are yet to be fully uncovered.

Here we present Cu-based double perovskite oxides ($A_2CuB'O_6$) with B-site rock-salt ordering and superexchange interaction to facilitate efficient and stable $CO_2$-to-$CH_4$ conversion. As the proof of concept, we employed $W^{6+}$ cations as the B' sites, mainly because of their low-lying unoccupied $5d$ states that strongly hybridized with O $2p$ states, and synthesized a double perovskite oxide of $Sr_2CuWO_6$ as the model catalyst for $CO_2RR$. As expected, for the $Sr_2CuWO_6$, its corner-linked octahedra of $CuO_6$ and $WO_6$ were rock-salt ordered. This unique structure made the nearest Cu cations very far apart from each other with a minimum distance of 5.4 Å and introduced superexchange interaction that was mainly manifested by O-anion-mediated electron transfer from Cu to W cations. When evaluated as a catalyst toward $CO_2RR$, relative to its physical-mixture counterpart and the reported Cu-based perovskite oxides, the $Sr_2CuWO_6$ delivered remarkable enhancements in activity and selectivity for $CH_4$, together with boosted stability. Our experiments and theoretical calculations suggested that such performance improvements were mainly attributed to the following aspects: the sufficiently long Cu–Cu distances promoting *CO hydrogenation but inhibiting C–C coupling; the superexchange interaction transferring the electrons (around Cu sites) to W sites during $CO_2RR$ and thus stabilizing the Cu sites (e.g., $Cu^+$).

## Results

### Crystal structure and long Cu–Cu distances

The $Sr_2CuWO_6$ catalyst was synthesized through a facile and scalable solid-state reaction (combined high-energy ball milling) process. Note that, according to the tolerance factor rule[10], another alkaline-earth metal cation, i.e., $Ba^{2+}$, can also be selected as the A-site cation to form a double perovskite of $Ba_2CuWO_6$. Since our work mainly focused on uncovering the key roles of superexchange-stabilized long-distance Cu sites in enhancing $CO_2RR$ property, either $Sr_2CuWO_6$ or $Ba_2CuWO_6$ can serve as the model catalyst in our work. For the proof of concept, here we designed and synthesized one of these two, i.e., $Sr_2CuWO_6$. The as-prepared sample had uniform particle size with an average value of around 300 nm together with a specific surface area of about 3 $m^2 g^{−1}$ (Supplementary Fig. 1). According to the inductively coupled plasma mass spectroscopy analysis, the chemical constituent of the $Sr_2CuWO_6$ sample was compatible with its nominal compositions (Supplementary Table 1). Figure 1a shows the X-ray diffraction (XRD) pattern and corresponding Rietveld refinement analysis (Supplementary Table 2) of the $Sr_2CuWO_6$ sample. The $Sr_2CuWO_6$ was characterized by a pure tetragonal B-site rock-salt-ordered double perovskite phase that was indexed to a space group of $I4/m$ with lattice parameters of $a = 5.436$ Å and $c = 8.400$ Å[35]. Here we also showed the crystal structure of $Sr_2CuWO_6$ in Fig. 1a. The structure consisted of alternating corner-sharing $WO_6$ and Jahn–Teller distorted $CuO_6$ octahedra (with short Cu–$O_{ab}$ bonds in the ab-plane and long Cu–$O_c$ bonds along the c-axis), with Sr cations situated at the void positions between these octahedra. As a result, the probably nearest Cu cations were far apart from each other, with two different distances of 5.4 and 5.7 Å that were induced by the Jahn–Teller distortion of $CuO_6$ octahedra[35] (Fig. 1b and Supplementary Fig. 2). These distances between adjacent Cu species were far enough to inhibit the C–C coupling and facilitate $CO_2$-to-$CH_4$ conversion, as to be discussed below.

We validated the crystal structure of $Sr_2CuWO_6$ using high-resolution transmission electron microscopy (HRTEM) and a selected-area electron-diffraction (SAED) pattern along the [1$\bar{1}$0] zone axis. In Fig. 1c–e, the tetragonal phase was observed, presenting clear crystal fringes with interplanar spacings of about 0.284 and 0.420 nm, corresponding to its (112) and (002) diffraction planes, respectively. Raman spectra further suggested the phase structure of $Sr_2CuWO_6$ crystallized with tetragonal $I4/m$ symmetry (Supplementary Fig. 3)[36]. The energy dispersive X-ray (EDX) mappings in Fig. 1f suggested the existence and homogeneous distribution of Sr, Cu, W, and O elements in the sample. Wide-scan X-ray photoelectron spectra (XPS) (Supplementary Fig. 4) also indicated that the sample was composed of the Sr, Cu, W, and O elements, without any detected signal of other elements except the reference C element.

### Superexchange interaction

We conducted XPS and synchrotron-based X-ray absorption spectra (XAS) to explore electronic structure information and superexchange interaction of $Sr_2CuWO_6$ catalyst. A physical mixture of $CuO/WO_3$ was prepared as a control sample (Supplementary Fig. 5), carrying the same molar ratio of Cu and W elements as the $Sr_2CuWO_6$. Figure 2a, b shows Cu $2p$ and W $4f$ spectra of the $Sr_2CuWO_6$. The peaks at 934.1 and 35.1 eV could be assigned to $Cu^{2+}$ $2p_{3/2}$ and $W^{6+}$ $4f_{7/2}$, respectively, illustrating the approximate valence states of Cu (+2) and W (+6) in $Sr_2CuWO_6$. Relative to the $CuO/WO_3$, the $Cu^{2+}$ $2p_{3/2}$ peak of $Sr_2CuWO_6$ shifted 0.33 eV to higher binding energy, whereas their $W^{6+}$ $4f_{7/2}$ peak underwent a negative shift of 0.26 eV. Such XPS peak shifts preliminarily suggest that there is electron redistribution (from $Cu^{2+}$ to $W^{6+}$) in the $Sr_2CuWO_6$.

Figure 2c, d shows the normalized Cu K-edge and W $L_3$-edge X-ray absorption near-edge structure spectra (XANES) for $Sr_2CuWO_6$. The absorption edges (i.e., Cu K-edge and W $L_3$-edge) of $Sr_2CuWO_6$ were nearly identical to those of CuO and $WO_3$ references, respectively,

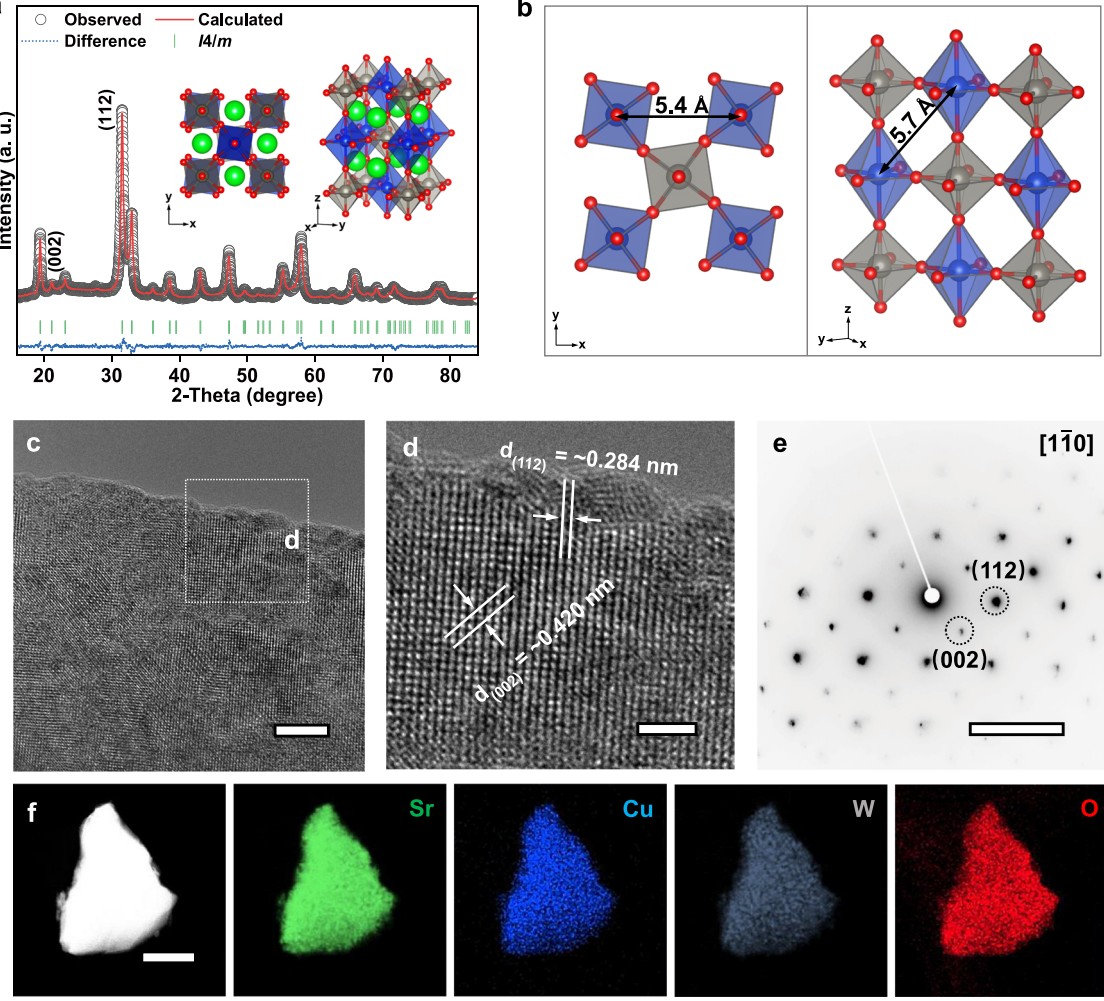

**Fig. 1 | Crystal structure and composition of Sr$_2$CuWO$_6$. a** Rietveld refinement plot of XRD data and schematic illustrations of crystal structure for Sr$_2$CuWO$_6$. Sr, Cu, W, and O are represented by green, blue, gray, and red dots, respectively. The blue and gray octahedra represent CuO$_6$ and WO$_6$ motifs, respectively. **b** Schematic illustrations of distances between the probably nearest Cu cations. **c** HRTEM image of Sr$_2$CuWO$_6$ (scale bar: 10 nm). **d** Enlarged HRTEM image of Sr$_2$CuWO$_6$ taken from the region marked in (**c**) (scale bar: 2 nm). **e** SAED pattern of Sr$_2$CuWO$_6$ (scale bar: 5 1/nm). **f** EDX mappings of Sr$_2$CuWO$_6$ (scale bar: 100 nm).

confirming the valence states of Cu and W species in Sr$_2$CuWO$_6$ close to +2 and +6. In the enlarged spectrum of the Cu K-edge (Fig. 2e), a positive-energy shift and higher white-line peak intensity were observed for the Sr$_2$CuWO$_6$, as compared to the CuO reference, indicative of the existence of a higher valence state of Cu species in Sr$_2$CuWO$_6$. On the contrary, the spectrum of the W L$_3$-edge for Sr$_2$CuWO$_6$ exhibited a slight shift towards lower energy and a weaker white-line peak intensity relative to the WO$_3$ reference (Fig. 2f), indicating a minor reduction of W valence state in Sr$_2$CuWO$_6$. These XAS results demonstrate electron interaction between CuO$_6$ and WO$_6$ octahedra or electron transfer in the direction from Cu to W species in the Sr$_2$CuWO$_6$. Combined with the above crystal structure characterization, one can believe that this electron transfer between rock-salt-ordered CuO$_6$ and WO$_6$ octahedra must be mediated by the intermediate oxygen anions, thus being defined as a superexchange interaction.

We further performed Bader charge analysis to investigate charge density redistribution. In Fig. 2g, the light-blue regions, surrounding the Cu, O, and W sites, clearly depicted the Cu−O−W charge transfer channels. The Bader charges of Cu and W sites in Sr$_2$CuWO$_6$ were calculated to be 1.27 and 2.95 |e|, respectively, which were different from 1.08 |e| for Cu sites in CuO and 3.08 |e| for W sites in WO$_3$ (Supplementary Table 3). These phenomena also indicate the charge redistribution from Cu to W sites (mediated by O sites) in Sr$_2$CuWO$_6$,

consistent with the XPS and XAS results. Thus, the B-site rock-salt-ordered double perovskite lattice was endowed with significant superexchange interaction (Cu−O−W) between alternate CuO$_6$ and WO$_6$ octahedra, mainly characterized by the O-anion-mediated electron transfer from Cu to W cations (Cu$^{2+}$ + W$^{6+}$ → Cu$^{>2+}$ + W$^{<6+}$), as schematically illustrated in Fig. 2h. As a result, we infer that the superexchange interaction could suppress the accumulation of electrode-supplied electrons around Cu sites via fast electron transport channels (light-blue regions in Fig. 2g), thereby protecting the Cu sites during CO$_2$RR. Besides, in light of the increased valence state (or electronegativity) of Cu sites reducing the electronegativity difference between Cu and O sites, the superexchange interaction could strengthen Cu−O bond covalency and thus maintain the Cu−O lattice integrity during CO$_2$RR. We proved the strengthened Cu−O bond covalency by the computed density of states (DOS) and band centers of Cu 3$d$ and O 2$p$ (Fig. 2i and Supplementary Fig. 6), using CuO as a reference.

## Activity and selectivity for CH$_4$

We carried out density functional theory (DFT) calculations to predict CO$_2$RR properties of Sr$_2$CuWO$_6$ catalyst (Fig. 3a, Supplementary Figs. 7 and 8). The DFT calculations were implemented on CuO$_2$/WO$_2$-terminated Sr$_2$CuWO$_6$(001) surface (Supplementary Fig. 7) since such a surface was usually observed and stable[37,38]. We took the full reaction

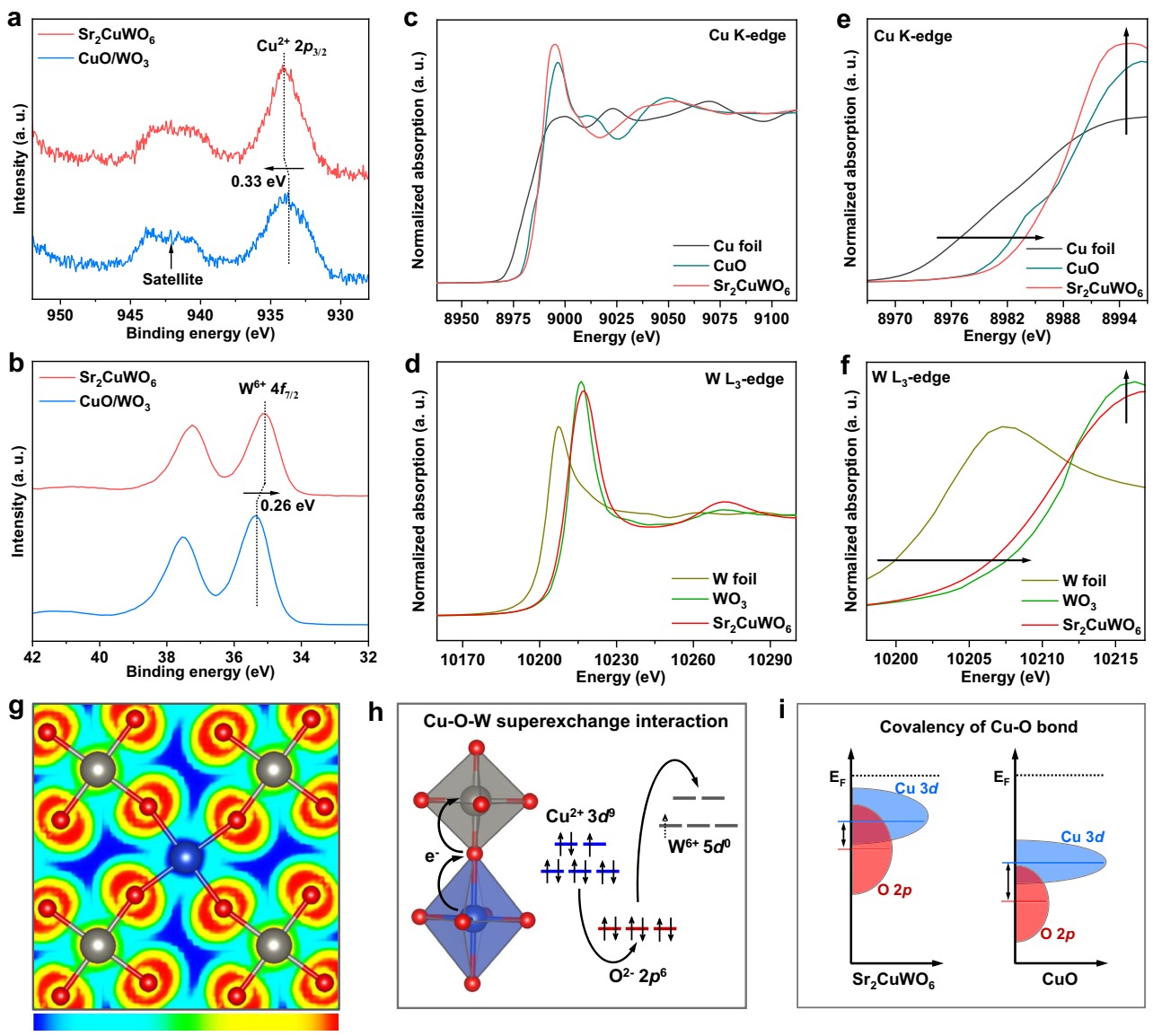

**Fig. 2 | Superexchange interaction in Sr₂CuWO₆.** **a** Cu 2*p* XPS spectra of Sr₂CuWO₆ and CuO/WO₃. **b** W 4*f* XPS spectra of Sr₂CuWO₆ and CuO/WO₃. **c** Cu K-edge XANES spectra of Sr₂CuWO₆. **d** W L₃-edge XANES spectra of Sr₂CuWO₆. **e** Enlargement of Cu K-edge XANES spectra. **f** Enlargement of W L₃-edge XANES spectra. **g** Top view of charge distribution between CuO₆ and WO₆ octahedra in Sr₂CuWO₆. Cu, W, and O are represented by blue, gray, and red dots, respectively. The light-blue regions (surrounding the Cu, O, and W sites) depict the electron transfer channels. **h** Schematic illustration of Cu−O−W superexchange interaction (electron transfer from Cu to W cations mediated by O anions) in Sr₂CuWO₆. The blue and gray octahedra represent CuO₆ and WO₆, respectively. **i** Schematic illustration of electronic DOS contributions (from O 2*p* and Cu 3*d* states) and Cu−O bond covalency for Sr₂CuWO₆ and CuO.

pathways for CH₄ and C₂H₄ formation starting from *CO as analysis objects and calculated their corresponding energy profiles at the Cu sites[39,40]. On the Sr₂CuWO₆(001) surface, the energy difference between *CO and *CHO was about 0.64 eV, much lower than the energy barrier (1.08 eV) for C₂H₄ production (i.e., 2*CO to the TS) (Fig. 3a and Supplementary Fig. 8). As a result, CH₄ formation was more favorable on the Sr₂CuWO₆(001) surface based on the presumption that the energy of TS for the *CO hydrogenation was not significantly different from the energy of *CO step. This could be ascribed to the fact that the long Cu−Cu distances (at least 5.4 Å) on Sr₂CuWO₆(001) surface were able to intensify the single-atomic feature of Cu, thereby inhibiting the C−C coupling but facilitating the CH₄ production. To this end, associated with its actual physicochemical properties, we can predict that the Sr₂CuWO₆ catalyst with B-site rock-salt-ordered structure will offer remarkable activity and selectivity toward CO₂-to-CH₄ conversion.

We preliminarily checked the probability of CO₂RR occurring over the Sr₂CuWO₆ catalyst by linear sweep voltammogram (LSV) curves recorded in a CO₂- and Ar-flowed liquid-electrolyte (1 M KOH) flow cell, respectively (Supplementary Figs. 9 and 10). Relative to Ar-flowed electrolyte, there were higher current densities as well as a less negative onset potential in CO₂-flowed electrolyte, suggesting that the Sr₂CuWO₆ catalyst is indeed active toward CO₂RR. We then systematically evaluated CO₂RR properties of the Sr₂CuWO₆ catalyst at various applied current densities in CO₂-flowed liquid-electrolyte (1 M KOH) flow cell (Fig. 3b, Supplementary Figs. 11 and 12). As a note, the 1 M KOH was adopted as the electrolyte in the flow cell because it was able to improve charge transfer, inhibit HER, and thus give rise to

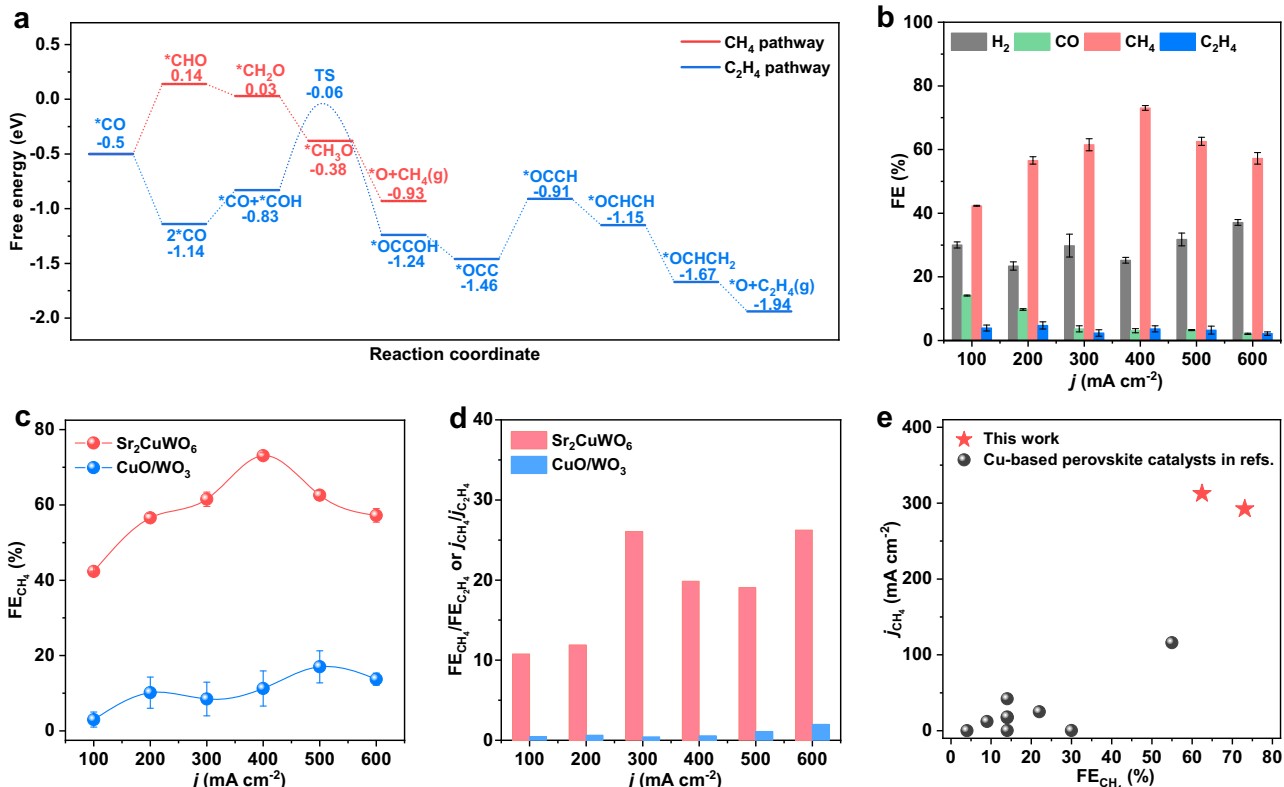

**Fig. 3 | Activity and selectivity for CH₄ over Sr₂CuWO₆. a** DFT-calculated energy diagrams for CH₄ and C₂H₄ formation on Sr₂CuWO₆(001) surface starting with *CO (TS: transition state). **b** FEs for various gas products over Sr₂CuWO₆ at different applied current densities. **c** FE$_{CH_4}$ of Sr₂CuWO₆ and CuO/WO₃ at different applied current densities. **d** FE$_{CH_4}$/FE$_{C_2H_4}$ or $j_{CH_4}$/$j_{C_2H_4}$ of Sr₂CuWO₆ and CuO/WO₃ at different applied current densities. **e** FE$_{CH_4}$ and $j_{CH_4}$ of Sr₂CuWO₆, in comparison with those of Cu-based perovskite oxides reported in the literature (Supplementary Table 4). The error bars represent the mean ± standard deviation (SD, $n$ = 3 replicates).

marked improvements in CO₂RR activity and selectivity, relative to the bicarbonate/carbonate electrolytes[13]. In the applied current density range (from 100 to 600 mA cm⁻²), the main product was CH₄, with high Faradaic efficiencies (FEs) more than 42.3% (Fig. 3b). At a current density of 400 mA cm⁻², the CH₄ product displayed a maximum FE of 73.1%, corresponding to a high partial current density of 292.4 mA cm⁻² exceeding the industrial-level requirements (>200 mA cm⁻²) (Fig. 3b and Supplementary Fig. 13). Meanwhile, the FE$_{C_2H_4}$ and FE$_{liquid\ C2+}$ ranged from 2.2% to 7.1% (Fig. 3b and Supplementary Fig. 12), indicating an efficient suppression of C–C coupling. These results reveal that upon serving as a catalyst toward CO₂RR, the Sr₂CuWO₆ is prone to generate CH₄ rather than C₂H₄, in line with the above DFT calculations (Fig. 3a).

We also benchmarked the CO₂RR properties of the Sr₂CuWO₆ against the CuO/WO₃. The detailed CO₂RR properties of the CuO/WO₃ were shown in Supplementary Fig. 14. The Sr₂CuWO₆ significantly promoted CO₂-to-CH₄ conversion, whereas its physical-mixture counterpart enhanced C–C coupling (similar to oxide-derived Cu catalysts[23,24]). To be specific, in the applied current density range, relative to the CuO/WO₃, the Sr₂CuWO₆ exhibited 3.7- to 14.1-fold higher FE$_{CH_4}$ or $j_{CH_4}$ (Fig. 3c and Supplementary Fig. 13), together with much lower FE$_{C_2H_4}$ or $j_{C_2H_4}$ (Supplementary Fig. 15). And the values (10.8–26.2) of FE$_{CH_4}$/FE$_{C_2H_4}$ or $j_{CH_4}$/$j_{C_2H_4}$ for the Sr₂CuWO₆ were almost 13.2–59.8 times higher than those (0.44–1.98) for the CuO/WO₃ (Fig. 3d). Combined with the above physicochemical property characterizations (Figs. 1 and 2) and DFT calculations (Fig. 3a), we could attribute these results to the sufficient-long Cu–Cu distances of Sr₂CuWO₆ that regulated the adsorption/activation of key intermediates, thus inhibiting C–C dimerization and promoting *CO hydrogenation. We compared the FE$_{CH_4}$ and $j_{CH_4}$ of Sr₂CuWO₆ catalyst

with those of the reported Cu-based perovskite oxides (Fig. 3e and Supplementary Table 4). The Sr₂CuWO₆ performed much better than all these perovskites reported in the literature. For instance, the $j_{CH_4}$ of Sr₂CuWO₆ was about 2.5–1562.5 times higher than that of the reported perovskite-based catalysts. To our knowledge, the Sr₂CuWO₆ was the most effective Cu-based-perovskite catalyst for CO₂-to-CH₄ conversion. Moreover, Supplementary Fig. 16 highlights that the activity and selectivity for CH₄ of Sr₂CuWO₆ are comparable to or higher than those of most reported representative Cu-based catalysts in flow cells (Supplementary Table 5).

## Cu sites stabilized by superexchange interaction

We performed a series of ex-situ and in-situ characterizations to investigate the structural evolution of Sr₂CuWO₆ and stabilization of Cu sites during CO₂RR (Fig. 4). The reduction tolerance of Sr₂CuWO₆ was probed under a high-temperature reducing atmosphere. At 300 °C (in H₂/Ar for 1 h), the Cu²⁺ in Sr₂CuWO₆ was reduced to Cu⁺, instead of metallic Cu (Fig. 4a and Supplementary Fig. 17), with the generation of oxygen vacancies (Supplementary Fig. 18 and Supplementary Table 6). Whereas the CuO/WO₃ was gradually reduced to Cu₂O/WO₃ (at 250 °C) and Cu/WO₃ (at 300 °C) (Supplementary Fig. 19). According to the Rietveld refinement analysis (Supplementary Fig. 20 and Supplementary Table 7), the Sr₂CuWO₆ underwent a phase transition from $I4/m$ to $Fm$-3$m$ during thermochemical reduction but still belonged to the category of B-site rock-salt-ordered double perovskites (Fig. 4a, b)[41]. This phase transition could be ascribed to lattice expansion of CuO₆ octahedra induced by reduction of smaller-size Cu²⁺ (0.87 Å) to larger-size Cu⁺ (0.91 Å). Notably, in the newly generated structure, the Cu–Cu distance (about 5.8 Å) was still very long (Fig. 4c and Supplementary Fig. 21), and the superexchange interaction (Cu–O–W) could still exist

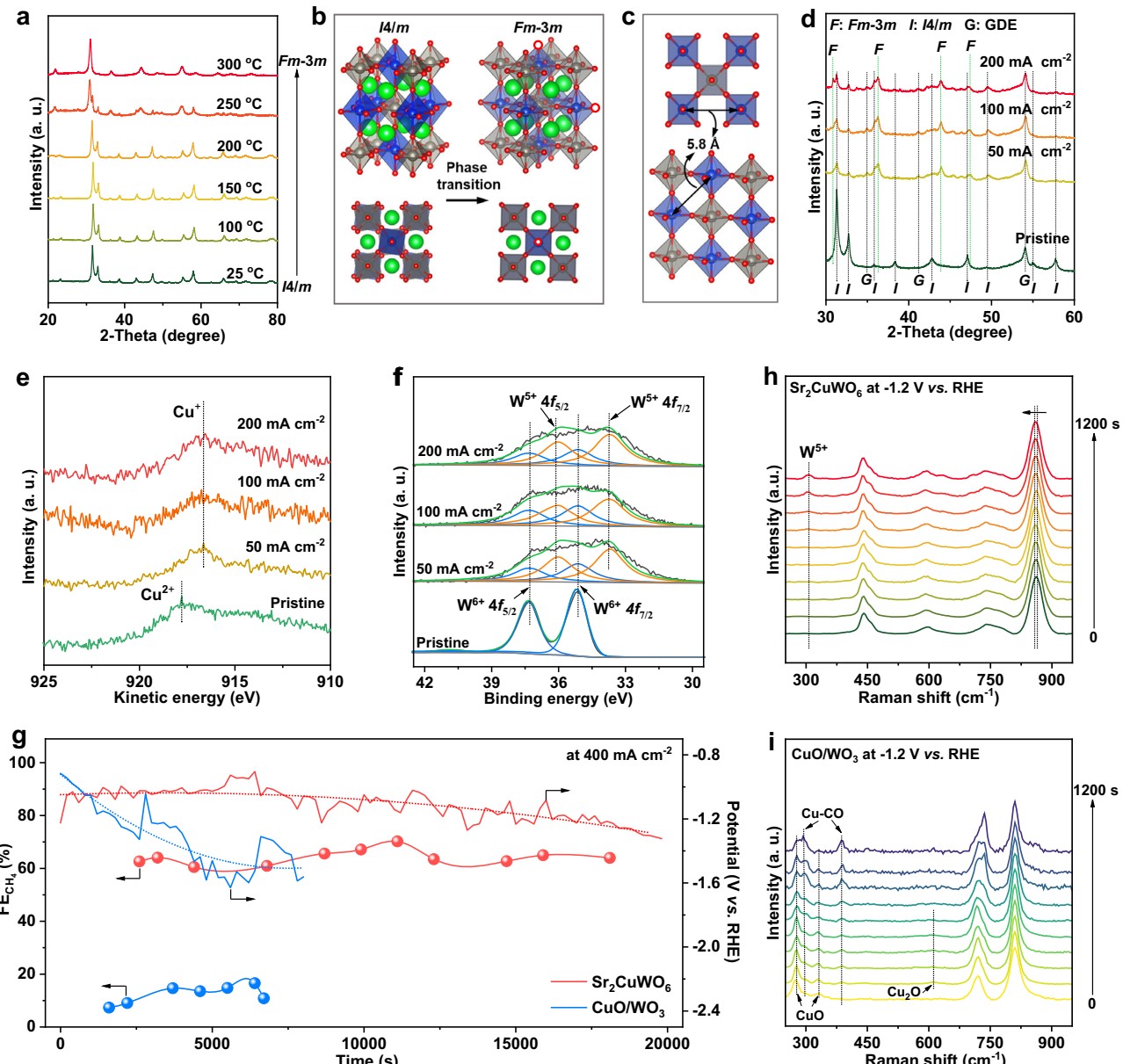

**Fig. 4 | Stability of the Cu sites. a** XRD pattern of $Sr_2CuWO_6$ after thermochemical reduction. **b** Schematic illustration of phase transition of $Sr_2CuWO_6$ after reduction. Sr, Cu, W, and O are represented by green, blue, gray, and red dots, respectively. The red-dotted circle, blue, and gray octahedra represent oxygen vacancy, $CuO_6$, and $WO_6$, respectively. **c** Schematic illustration of the distances between the nearest Cu cations in *Fm*-3*m* phase of $Sr_2CuWO_6$. **d** XRD patterns of $Sr_2CuWO_6$ after CO$_2$RR at different current densities (GDE: gas diffusion layer). **e** Cu LMM XPS spectra of $Sr_2CuWO_6$ after CO$_2$RR at different current densities. **f** W XPS spectra of $Sr_2CuWO_6$ after different current densities. **g** CO$_2$RR stability test of $Sr_2CuWO_6$ and $CuO/WO_3$ in a flow cell at 400 mA cm$^{-2}$. In-situ Raman spectra of **h** $Sr_2CuWO_6$ and **i** $CuO/WO_3$ as a function of CO$_2$RR time at −1.2 V vs. RHE (RHE: reversible hydrogen electrode).

or even be strengthened due to the easier electron transfer from Cu$^+$ to W$^{6+}$ sites relative to that from Cu$^{2+}$ to W$^{6+}$ sites (Supplementary Fig. 22). These results indicate that the superexchange interaction can inhibit deep reduction of the Cu sites and thus avoid structural collapse of the $Sr_2CuWO_6$. These may partly imply the high structural stability of $Sr_2CuWO_6$ during CO$_2$RR.

The possible structural changes of $Sr_2CuWO_6$ after CO$_2$RR were analyzed using ex-situ XRD and XPS (Fig. 4d–f). Similar to the thermochemical reduction, after CO$_2$RR (e.g., at 200 mA cm$^{-2}$), part of the *I*4/*m* phase of $Sr_2CuWO_6$ was converted into *Fm*-3*m* phase (Fig. 4d), without detectable impurity, and the Cu$^{2+}$ and part W$^{6+}$ species on the surface were reduced to Cu$^+$ and W$^{5+}$, respectively (Fig. 4e, f, Supplementary Fig. 23, and Supplementary Table 8). This suggests that the Cu$^+$ species (in the *Fm*-3*m* phase) might be active sites for CO$_2$

methanation. By contrast, as fully evidenced by previous studies, the CuO (in $CuO/WO_3$) can completely be reduced to metallic Cu under similar CO$_2$RR conditions[26,27]. As a result, the Cu sites of oxidation states in double perovskite structure are well stabilized by the superexchange interaction during CO$_2$RR. On this basis, we evaluated CO$_2$RR stability of the $Sr_2CuWO_6$ in comparison with the $CuO/WO_3$ through chronopotentiometric polarization in the CO$_2$-flowed liquid-electrolyte flow cell (Fig. 4g). During 20,000 s of electrolysis (at 400 mA cm$^{-2}$), for the $Sr_2CuWO_6$, the applied potential was stable at 1.23 ± 0.15 V vs. RHE, and the FE$_{CH_4}$ was maintained at 64% ± 6%. Whereas the $CuO/WO_3$ showed severe deterioration in potentials and obvious fluctuations in FE$_{CH_4}$ during 8000 s of electrolysis. These results demonstrate excellent CO$_2$RR stability of the $Sr_2CuWO_6$. As a note, our gas diffusion layer suffered flooding issues when the CO$_2$RR

stability test of $Sr_2CuWO_6$ catalyst ran for more than 20,000 s. Since the flooding issues can cause an essential failure of the $CO_2RR$ system, we terminated the electrolysis on the $Sr_2CuWO_6$ catalyst at about 20,000 s; but the steady $CO_2RR$ testing time for the $Sr_2CuWO_6$ catalyst itself was supposed to be for more than 20,000 s by considering the well-stabilized Cu sites. Usually, the flooding issues can be mitigated by washing away the carbonate precipitation. This has been widely used to reactivate the electrodes[42,43].

Moreover, the stability of Cu sites in $CuO_6$ octahedra of $Sr_2CuWO_6$ during $CO_2RR$ was further verified by in-situ Raman spectroscopy in an operando electrolyzer (Supplementary Fig. 24). As expected, the in-situ Raman spectroscopic analyses were consistent with the above ex-situ characterizations. To be specific, as the applied potential negatively shifted (from −0.8 to −1.2 V vs. RHE), the Raman spectra of $Sr_2CuWO_6$ displayed no change in characteristic peaks and no formation of any new peak (Supplementary Fig. 25). In addition, these characteristic peaks were also retained during 1200 s of electrolysis at −1.2 V vs. RHE (Fig. 4h). Note that, within the electrolysis time, only a new peak at about 308.2 $cm^{-1}$ appeared (Fig. 4h), possibly originating from the electrochemical reduction of $W^{6+}$ to $W^{5+}$[44]. It was also observed that a main characteristic peak at 864.1 $cm^{-1}$ gradually moved to lower Raman shifts (Fig. 4h), probably corresponding to the reduction-induced lattice expansion and phase transition (from $I4/m$ to $Fm\text{-}3m$) as mentioned above[36]. However, for the $CuO/WO_3$ during 1200 s of electrolysis (Fig. 4i), the characteristic peaks at 278.1 and 331.2 $cm^{-1}$ of CuO gradually disappeared, while a characteristic peak at 613.6 $cm^{-1}$ of $Cu_2O$ appeared and then disappeared, and the intensity of Cu−CO peaks at 294.2 and 385.6 $cm^{-1}$ was gradually improved[45]. These indicate that the CuO in $CuO/WO_3$ is initially reduced to $Cu_2O$ and then to metallic Cu.

In light of the above analyses, we plotted out the structural evolution diagrams to graphically describe the key role of superexchange interaction in stabilizing the Cu sites in $Sr_2CuWO_6$ during $CO_2RR$ (Fig. 5). As shown in Fig. 5a, once the $CO_2RR$ was initiated, the $Cu^{2+}$ on $Sr_2CuWO_6$ surface began to be reduced to $Cu^+$ by the electrode-supplied electrons. At the same time, the $Sr_2CuWO_6$ surface was transformed from the $I4/m$ to $Fm\text{-}3m$ phase, still being the B-site rock-salt ordered structure. Further reduction was not able to convert the $Cu^+$ to $Cu^0$ but rather led to the transformation of $W^{6+}$ to $W^{5+}$ on the surface while maintaining the $Fm\text{-}3m$ phase. The most plausible reason was that, during $CO_2RR$, the superexchange interaction effectively

transferred the electrode-supplied excessive electrons accumulated around the $Cu^+$ sites to $W^{6+}$ sites (to form $W^{5+}$) through the fast electron transport channels (Fig. 2g), thereby protecting the Cu sites from electron attack and preserving the double perovskite phase. By contrast, the $CuO/WO_3$ without superexchange interaction was successively reduced to $Cu_2O/WO_3$ and $Cu/WO_3$ (Fig. 5b). Taken together, during $CO_2RR$, although these changes occurred on the $Sr_2CuWO_6$ surface, the superexchange interaction prevented structural collapse, stabilized the $Cu^+$ sites, and maintained the long Cu−Cu distances, thereby promoting the efficient and stable $CO_2$-to-$CH_4$ conversion.

## Discussion

Employing $Sr_2CuWO_6$ as the proof-of-concept catalyst, we have developed Cu-based rock-salt-ordered double perovskite oxides for efficient and stable $CO_2$-to-$CH_4$ conversion and uncovered the key roles of their unique physicochemical properties in boosting activity, selectivity, and stability toward $CH_4$ production. In the rock-salt-ordered structure, the corner-linked $CuO_6$ and $WO_6$ octahedra alternated in all three crystallographic dimensions, leading to sufficiently long Cu−Cu distances (at least 5.4 Å) and marked Cu−O−W superexchange interaction. When explored as a catalyst toward $CO_2RR$, relative to its physical-mixture counterpart, the $Sr_2CuWO_6$ featured not only enhancements in terms of activity and selectivity for $CH_4$ but also significantly boosted stability. Moreover, the $Sr_2CuWO_6$ was the most effective Cu-based-perovskite catalyst for $CO_2$ methanation and performed comparably to or better than most reported representative Cu-based catalysts. According to the experiments and theoretical calculations, the superb performance could be attributed to the following factors: (i) the long-distance Cu sites facilitating *CO hydrogenation while inhibiting C−C coupling; (ii) the superexchange interaction stabilizing the Cu sites and preventing structural collapse. This work discovered efficient and stable Cu-based double perovskite oxides for $CO_2RR$, providing a new avenue for the rational design of more advanced Cu-based catalysts.

## Methods
### Chemicals and materials
All chemicals were used directly without any further purification. $SrCO_3$ (AR, ≥99%) and isopropanol (AR, ≥99.7%) were purchased from Sinopharm Chemical Reagent Co., Ltd. Dimethyl sulfoxide (DMSO,

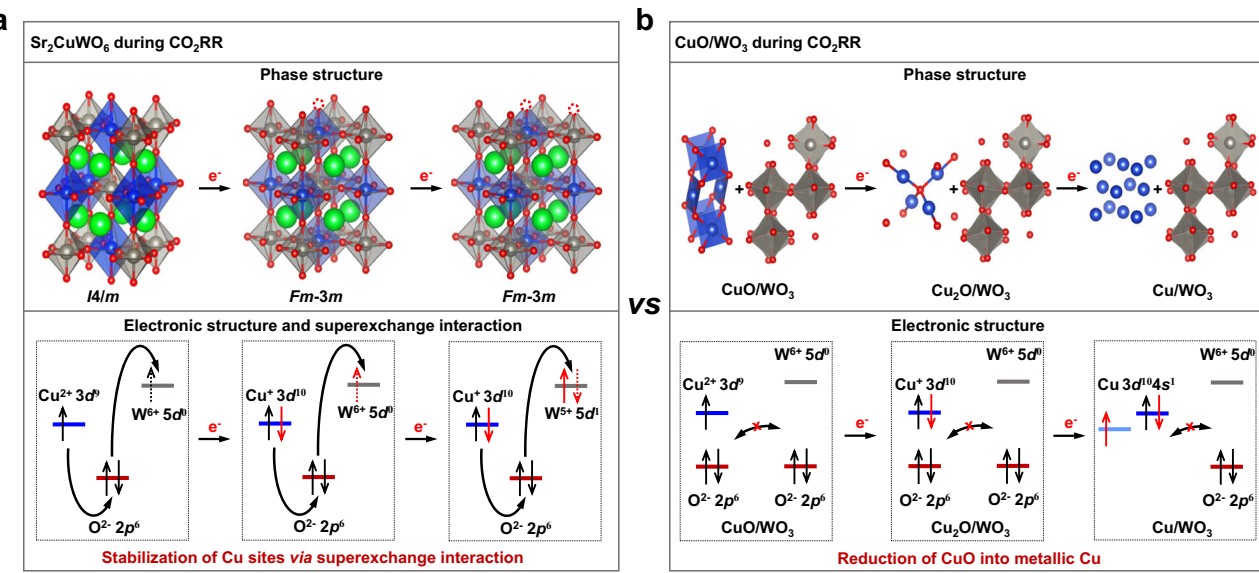

**Fig. 5 | Schematic illustrations of catalyst structure evolution during $CO_2RR$. a** $Sr_2CuWO_6$. **b** $CuO/WO_3$. Sr, Cu, W, and O are represented by green, blue, gray, and red dots, respectively. The red dotted circle represents oxygen vacancy.

≥99.9%) was purchased from Shanghai Macklin Biochemical Co., Ltd. CuO (AR, 99%) and $WO_3$ (AR, 99.8%) were purchased from Shanghai Aladdin Biochemical Technology. Nafion 117 solution (5 wt%) and $D_2O$ (99.9 atom% D) were purchased from Sigma-Aldrich Biochemical Technology. $HNO_3$ (AR, 65–68%), $H_2O_2$ (AR, 30%), HF (AR, ≥40%), and HCl (AR, 36–38%) were purchased from Xilong Scientific. High-purity $CO_2$ gas (99.999%), Ar gas (99.999%), and 10 vol% $H_2$–Ar gas (99.999%) were purchased from Qingdao Dehaiweiye Technology Co., Ltd.

## Synthesis
$Sr_2CuWO_6$ was synthesized by a high-temperature solid-state reaction/high-energy ball-milling process. In a typical procedure, stoichiometric $SrCO_3$, CuO, and $WO_3$ were well mixed by ball-milling process and then calcined at 900 °C in Air for 12 h. The admixture was ground again and then pressed into pellets under the pressure of 10 MPa for subsequent re-calcination at 1050 °C in Air for 24 h. Finally, the as-prepared powder was ground with the high-energy ball-milling process (900 rpm) to obtain uniform nanoparticles. The high-temperature reducing-atmosphere treatments of $Sr_2CuWO_6$ were processed in a sealed tube furnace in 10 vol% $H_2$–Ar mixture with a flow rate of 20 mL min⁻¹ for 1 h.

## Theoretical calculations
First-principles calculations were carried out on the basis of periodic density functional theory (DFT) using a generalized gradient approximation within the Perdew–Burke–Ernzerh of exchange correction functional with Vienna ab initio simulation package (VASP)[46,47]. Geometry optimization was conducted in $Sr_2CuWO_6$, CuO, and $WO_3$. The wave functions were constructed from the expansion of plane waves with an energy cutoff of 450 eV. Gamma-centered $k$-point of $3 \times 3 \times 1$ has been used. The consistence tolerances for the geometry optimization were set as $1.0 \times 10^{-6}$ eV/atom for total energy and 0.02 eV/Å for force, respectively. In order to avoid the interaction between the two surfaces, a large vacuum gap of 15 Å has been selected in the periodically repeated slabs. Static calculations were conducted with a convergence condition of $1.0 \times 10^{-6}$ eV for density of state (DOS), Bader charge, and electron localization function analysis. The band center of Cu $3d$ or O $2p$ was calculated using the following equation[48]:

$$E_t = \frac{\int_{-\infty}^{\infty} E \cdot T(E) dE}{\int_{-\infty}^{\infty} T(E) dE}$$

where $T(E)$ is the density of states (DOS) of orbitals. $E$ corresponds to the occupied state ranges below the fermi energy level ($E_F$) in DOS. Climbing image nudged elastic band (CI-NEB) was used for transition state searching. In free energy calculations, the entropic corrections and zero-point energy (ZPE) have been included. The free energy of species was calculated according to the standard formula:

$$\Delta G = E + \Delta ZPE + \Delta H - \Delta TS$$

where ZPE is the zero-point energy, $\Delta H$ is the integrated heat capacity, $T$ is the temperature of the product, and $S$ is the entropy.

## Characterization
X-ray diffraction (XRD) patterns were recorded by Rigaku Miniflex 600 (Hitachi) diffractometer with Cu Kα radiation (1.5418 Å). The Rietveld refinements of obtained data were conducted using FullProf software. Scanning electron microscopy (SEM) images were taken by a Hitachi S4800 microscope. Transmission electron microscopy (TEM) images were taken by a JEOL 2010F microscope (operated at 200 kV). To further confirm the structure and elements distribution, high-resolution TEM (HRTEM) and energy dispersive X-ray (EDX) spectra/mappings were performed on a JEOL ARM 300 F microscope equipped with dual EDX detectors. X-ray photoelectron spectroscopy (XPS) analyses were carried out by the Thermo ESCALAB 250Xi spectrometer with monochromated Al Kα radiation ($hv = 1486.6$ eV) operating at 150 W. The energies of each element were calibrated by the adventitious C1s (284.8 eV). Raman spectra were performed on a Renishaw Qontor spectrometer equipped with a 532 nm laser beam and a ×63 water-immersion objective lens. X-ray absorption spectroscopy (XAS) of Cu K-edge and W $L_3$-edge were obtained in a Singapore synchrotron light source (SSLS), using an XAFCA Beamline (operated at 700 MeV) with a maximum current of 200 mA. The reference samples, such as CuO, Cu foil, $WO_3$, and W foil were also measured for comparison and energy calibration. All XANES data were measured in transmission mode using an ion chamber detector with a Si 111 monochromator and analyzed by the Athena program[49]. The nitrogen adsorption and desorption processes were recorded on an Autosorb-iO (Quantachrome) device at the boiling point of liquid nitrogen to calculate the specific surface areas by the Brunauer–Emmett–Teller (BET) method. The inductively coupled plasma mass spectrometer (ICP–MS) (Agilent 730) was applied to test the metal contents of $Sr_2CuWO_6$. The samples for ICP–MS analysis were obtained by dissolving 10 mg sample powder with the mixture of 5 mL $HNO_3$, 1 mL $H_2O_2$, 1 mL HCl, and 0.5 mL HF in the oven at 180 °C for 8 h. The cooled-down solution was further diluted to a level of 100 ppb by using a 1% $HNO_3$ solution.

## Preparation of working electrodes
The working electrodes were prepared by coating the catalyst ink onto the hydrophobic carbon paper (i.e. the gas diffusion layer, GDL). To be specific, for the preparation of the catalyst ink, 10 mg sample powder was homogenously dispersed into a mixed solution of isopropanol (1 mL) and Nafion (50 μL) by ultrasonic processing for 1 h. The catalyst ink was then coated on the hydrophobic carbon paper (Toray, YLS-30T, $1.5 \times 1.5$ cm²) with a loading amount of 0.5 mg cm⁻² and dried under the infrared lamp. This method was used to prepare the working electrodes for both electrochemical measurements and the in-situ Raman spectroscopic measurements.

## Electrochemical measurement
The $CO_2$ electrochemical reduction measurements were processed in a homemade flow cell with a three-electrode system controlled by a CS310M electrochemical workstation (Wuhan, Corrtest). The Ag/AgCl electrode (filled with saturated KCl solution) and Pt mesh were used as the reference and counter electrodes, respectively. 1 M KOH was used as the electrolyte, filling, and cycling in the flow cell with a pumped rate of 20 mL min⁻¹ controlled by a double channel peristaltic pump (Hebei, Leadfluid, BQ80s). An anion-exchange membrane (Hefei, ChemJoy Polymer Materials Co., Ltd., SYMA-2) was used for separating the anodic and cathodic compartments to avoid crossover pollution. High-purity $CO_2$ gas was continuously supplied into the gas chamber with a flow rate of 35 mL min⁻¹ controlled by a mass flow controller (D07-19B, Sevenstar Electronics Co., Ltd, Beijing) and the flow rate was further verified by a soap bubble flowmeter. The LSV curves were also recorded in the flow-cell configuration flowed with Ar or $CO_2$ gas at a scan rate of 10 mV s⁻¹. All applied potentials were converted into the standard reversible hydrogen electrode (RHE) potentials by the equation of $E_{RHE} = E_{Ag/AgCl} + 0.197 \text{ V} + 0.0591 \text{ V} \times pH$, with 70% iR compensation. The cell resistance was measured using the function of $R$s measurement in the measurement soft of Corrtest CS310M electrochemical workstation (the value at 10,000 HZ from the electrochemical impedance spectroscopy) under open circuit potentials before every independence test.

## Quantification of products
The gas products were detected by online gas chromatography (GC2060, Ramiin, Shanghai) equipped with flame ionization (FID) and thermal conductivity (TCD) detectors. A standard gas mixture (containing 1 vol% each of $H_2$, CO, $CH_4$, $C_2H_4$, $C_2H_6$, and 95 vol% $CO_2$) was used to calibrate the gas products. The Faradaic efficiency (FE) of each gas product under different current densities was gained based on

more than three parallel experiments. After the reaction, the catholyte was collected for liquid product analyses by a Nuclear magnetic resonance spectrometer (NMR, Bruker, AVANCE-III 600 Hz). Typically, 2 mL catholyte was mixed with 100 μL 5 mM DMSO (as an internal standard substance). And then 250 μL mixture was mixed with 350 μL $D_2O$ for the NMR measurement. The FEs of the products were calculated by the following equation:

$$FE = \frac{Q_{product}}{Q_{total}} = \frac{n \cdot N \cdot F}{Q_{total}}$$

where $Q_{product}$ and $Q_{total}$ present the charge consumption of the target product and $CO_2RR$ process, respectively, $n$ presents the electron transfer number of the target product, $N$ presents the amount of substance for the product and can be calculated from the product concentration, and $F$ presents the faradaic content (96,485 C mol$^{-1}$).

## In-situ Raman test

The in-situ Raman test was processed in an electrochemical operando cell (C031-2, Tianjin Gaoss Union Technology Co. Ltd.) with a three-electrode system. The in-situ Raman spectra were recorded by the Renishaw Qontor spectrometer using a 532 nm laser beam and a ×63 water-immersion objective lens. The Ag/AgCl (filled with saturated KCl solution) and graphite electrodes were used as reference and counter electrodes, respectively. The carbon paper coated with catalyst ink was used as the cathode, immersed in the $CO_2$-saturated 0.1 M $KHCO_3$ electrolyte. The 0.1 M $KHCO_3$ was filled in the cathodic compartment while flowing in the anodic compartment with a flow rate of 20 mL min$^{-1}$ to remove bubbles. The water-immersion objective lens was immersed in the cathodic compartment to directly observe the surface of the catalyst. The cathodic and anodic compartments were separated by a Nafion 117 proton exchange membrane. The power of the laser was kept at 1 mW to avoid irradiation damage on the catalyst. The surface exposure time was 20 s, and each signal line was collected twice. All Raman raw data were recorded and processed by Wire soft.

## Data availability

All data generated in this study are provided in the article and Supplementary Information. Additional data related to this study are available from the corresponding authors on request. The raw data generated in this study are provided in the Source Data file. Source data are provided with this paper.

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

## Acknowledgements
J.W.Z. acknowledges the support from the National Natural Science Foundation of China (52102258), the Taishan Scholars Program (tsqn202306309), the Natural Science Foundation of Shandong Province (ZR2023YQ012), and the Natural Science Foundation of Jiangsu Province (BK20210447). X.W. acknowledges the support from the ECS grant from the Research Grants Council of the Hong Kong Special Administrative Region (Project No. 21300323) and the CityU start-up fund (Project No. 9610600).

## Author contributions
J.W.Z. and H.Q.J. conceived the research. Y. Zhang prepared the catalysts and performed electrocatalysis, characterizations, and theoretical calculations. Z.T.C., X.Z.T., and X.D.B. carried out TEM measurements and analyses. Z.B.Z. conducted XRD analyses. Y. Zhang and M.H.H. performed in-situ Raman tests and analyses. J.W.Z., Y. Zhang, M.H.H., X.W., and Y.F. Zhu wrote and revised the manuscript. All authors added to the discussion and contributed to the preparation of the manuscript. J.W.Z. and H.Q.J. supervised the work.

## Competing interests
The authors declare no competing interests.
