## [Peer Review File · Nature Communications]

REVIEWER COMMENTS

Reviewer #1 (Remarks to the Author):

Zhu et al developed a new Cu-based rock-salt-ordered double perovskite oxide of Sr₂CuWO₆ for efficient and stable CO₂ electromethanation. The unique rock-salt-ordered structure of Sr₂CuWO₆ introduced long Cu-Cu distance and Cu-O-W superexchange interaction which effectively suppressed C-C coupling as well as induced charge rearrangement to protect the active sites, resulting in enhanced CO₂ methanation performance. This manuscript is well structured with sufficient supporting data, presenting the good potential of double perovskite in CO₂RR. This innovative design will bring broad interests to the application of perovskite-based catalysts in CO₂RR and open up a new promising way for designing high-performance CO₂RR catalysts. I would like to recommend its publication after minor revisions.

1. The reason for choosing Sr as A-site is not fully discussed. Please elaborate on the reasons for using Sr rather than Ba.
2. In this manuscript, 1 M KOH was adopted as the electrolyte. However, the CO₂ will react with KOH and form the carbonate precipitates, which would lead to membrane blocking and even further decrease the catalysts performance. Please distinguish this electrolyte selection from widely adopted carbonate electrolyte.
3. As flooding issue is inevitable in flow cell, the authors should account for the influence of flooding on catalyst performance and how to mitigate it.
4. How to derive the band center of Cu and O? Please add relevant information.
5. The intermediates adsorption models of TS state and *COCOH step in Fig 3a may be wrong, which should be exchanged.

Reviewer #2 (Remarks to the Author):

In the manuscript entitled " Superexchange-Stabilized Long-Distance Cu Sites in Rock-Salt-Ordered Double Perovskite Oxides for CO₂ Electromethanation," the authors assert that the superexchange interaction contributes to enhancing the stability of oxide perovskite electrocatalysts. They propose that the superexchange interaction facilitates electron transfer from Cu to W, thereby reducing the electron density at the Cu site and modulating catalytic activity and selectivity. To support these claims, the authors present XPS and XANES spectra in conjunction with the results of density functional theory (DFT) calculations.

However, I am hesitant to conclude that there exists clear experimental or computational evidence establishing the superexchange interaction as a pivotal factor during catalysis. The only aspect that can be unequivocally supported by experimental and theoretical data is that the Cu site in Sr₂CuWO₆ exhibits lower electron density than CuO. Nevertheless, there is no direct evidence demonstrating whether this reduction is attributable to the superexchange interaction or how it might influence changes in catalytic performance. It is imperative that we distinguish between correlation and causation.

Consequently, I find the present manuscript to be premature, and I do not support its publication. My detailed comments are provided below:

1. (Page 3, lines 55-56) The statement that Cu-based perovskite oxides could be active towards CO₂ electromethanation by partially or entirely occupying the B sites is highly speculative. This argument lacks a solid logical foundation and lacks proper citations to substantiate this claim.
2. (page 3, line 71-73) "The formation of double perovskite ... in CH₄ production.": This claim lacks a clear basis and appears overly speculative.
3. (page 4, line 80-81) "These superexchange interaction may hinder electron accumulation around the Cu sites, ... , thereby boosting the catalytic activity during CO₂RR.": To the best of my knowledge, the superexchange interaction primarily plays a role in determining the spin arrangements within the system and is not strongly associated with the degree of electron accumulation. Furthermore, there is no established mechanistic link between the superexchange interaction and catalytic activity.
4. (page 7, line 168) The authors argued that the distinct partial charge distributions at the Cu and W sites of Sr₂CuWO₆, as compared to CuO+WO₃, are mediated by "O sites. In order to substantiate the existence of a superexchange interaction, it is crucial to demonstrate the involvement of the 2p orbitals of the O sites. However, there is currently no compelling evidence presented to establish that O atoms indeed mediate the charge redistribution as claimed.
5. (Figure 3a) In Figure 3a, the authors juxtapose the reaction energy difference for the CH₄ production path with the barrier energy for the ethylene production path in order to draw a comparison. However, this approach represents a biased comparison and does not provide a comprehensive assessment of the energetics for the entire reaction pathways. A more rigorous analysis would involve comparing the energetic profiles of the full reaction paths for both CH₄ and ethylene production.

Reviewer #3 (Remarks to the Author):

Jiawei Zhu & Heqing Jiang et al present a Cu-based double perovskite oxides A₂CuB'O₆ with B-site rock-salt ordering to boost CO₂-to-CH₄ conversion, inhibiting C-C coupling reactions. As model catalyst, the authors synthesised Sr₂CuWO₆, exhibiting corner-linked CuO₆ with high Cu-Cu distances. Indeed, the performance of the systems for CO₂RR is remarkable with a ~73% FE and -400mA/cm², which is a competitive number, but does not stand out in the field.

The reviewer noted the remarkably long-term stability of the Sr₂CuWO₆ at reductive potentials. The authors proposed that high stability stems from a super exchange stabilisation of the Cu sites, proposed via O-anion-mediated ET from Cu to W₆ cations. This reviewer is not found of this explanation as other factors were not considered. However, the presented calculations and rationale do support the claims of the authors. In situ and ex situ spectroscopic experiments support that the material does not undergo dramatic transformation, which is unique.

It should be noted that this reviewer cannot in detail comment on the DFT calculations performed. This reviewer noted that methods and approaches (Bader charge calculations etc.) employed presented in this manuscript are in line with other works in this field.

The manuscript is clearly outlined and very well written. Synthesis of the Sr₂CuWO₆ is well described. The structural characterisation has been carried out with care and seems to verify the structure of the proposed Sr₂CuWO₆ the structure. The spectroscopic experiments are well presented.

This reviewer believes that the work and the basics of the research is original, and the performance, i.e., high FE and j(CH₄), long-term stability, is competitive. Thus, the work will be of interest for those working in the field of electrocatalysis, perovskites, and CO₂RR, and beyond.

Rebuttal/Response Letter to Reviewers

Reviewer #1

“Zhu et al developed a new Cu-based rock-salt-ordered double perovskite oxide of Sr₂CuWO₆ for efficient and stable CO₂ electromethanation. The unique rock-salt-ordered structure of Sr₂CuWO₆ introduced long Cu-Cu distance and Cu-O-W superexchange interaction which effectively suppressed C-C coupling as well as induced charge rearrangement to protect the active sites, resulting in enhanced CO₂ methanation performance. This manuscript is well structured with sufficient supporting data, presenting the good potential of double perovskite in CO₂RR. This innovative design will bring broad interests to the application of perovskite-based catalysts in CO₂RR and open up a new promising way for designing high-performance CO₂RR catalysts. I would like to recommend its publication after minor revisions.”

Response: We thank the reviewer for the constructive comments and suggestions. All the suggestions and comments have been carefully addressed and revisions have been made accordingly.

1. *“The reason for choosing Sr as A-site is not fully discussed. Please elaborate on the reasons for using Sr rather than Ba.”*

Response: We thank the reviewer for raising a good point. According to the tolerance factor rule (*Chem. Rev.* **2015**, *115*, 9869), both Sr²⁺ and Ba²⁺ cations are suitable for the formation of Cu/W-based double perovskite oxides, such as Sr₂CuWO₆ and Ba₂CuWO₆ (*J. Solid State Chem.* **2006**, *179*, 3556). In the present work, we mainly focused on uncovering the respective roles of the long Cu-Cu distance and Cu-O-W superexchange interaction in boosting activity/selectivity and stability toward CO₂ methanation, rather than investigating the influence of A-site cations on CO₂RR property. Therefore, as it was feasible to choose either Sr²⁺ or Ba²⁺, for the proof of concept, we designed and synthesized the Sr₂CuWO₆ catalyst in our work. We have added a brief note about this point into the revised manuscript (page 4-5), as shown below.

Page 4-5, line 106-111:

“Note that, according to the tolerance factor rule¹⁰, another alkaline-earth metal cation, *i.e.*, Ba²⁺, can also be selected as the A-site cation to form a double perovskite of Ba₂CuWO₆. Since our work mainly focused on uncovering the key roles of superexchange-stabilized long-distance Cu sites in enhancing CO₂RR property, either Sr₂CuWO₆ or Ba₂CuWO₆ can serve as the model catalyst in our work. For the proof of concept, here we designed and synthesized one of these two, *i.e.*, Sr₂CuWO₆.

10. Chen, D., Chen, C., Baiyee, Z. M., Shao, Z. & Ciucci, F. Nonstoichiometric oxides as low-cost and highly-efficient oxygen reduction/evolution catalysts for low-temperature electrochemical devices. *Chem. Rev.* **115**, 9869-9921 (2015).”

2. *“In this manuscript, 1 M KOH was adopted as the electrolyte. However, the CO₂ will react with KOH and form the carbonate precipitates, which would lead to membrane blocking and even further decrease the catalysts performance. Please distinguish this electrolyte selection from*

widely adopted carbonate electrolyte.”

Response: We thank the reviewer for raising an interesting point. As the reviewer pointed out correctly, in flow cell, when 1 M KOH is selected as the electrolyte, some bicarbonate/carbonate would be formed and even precipitated on the cathode and membrane, leading to catalytic performance degradation. In spite of this, in general, according to many representative literatures (*e.g.*, *Nat. Catal.* **2020**, *3*, 478; *Nat. Catal.* **2022**, *5*, 1081; *Angew. Chem. Int. Ed.* **2021**, *60*, 466; *J. Am. Chem. Soc.* **2021**, *143*, 3808), highly alkaline electrolytes (*e.g.*, 1 M KOH) are frequently used in the flow cell for fundamental research. The main reason is that, relative to the bicarbonate/carbonate electrolytes (*e.g.*, 1 M KHCO₃), the highly alkaline electrolytes have been proved to be able to improve charge transfer kinetics, suppress the competitive HER, and thus give rise to marked improvements in CO₂RR activity and selectivity (*Adv. Mater.* **2022**, *34*, 2206002). Hence, based on these advantages, the 1 M KOH was selected as the electrolyte in the present work. We have added the related discussion into the revised manuscript (page 9), as shown below.

Page 9, line 226-228:

“As a note, the 1 M KOH was adopted as the electrolyte in flow cell because it was able to improve charge transfer, inhibit HER, and thus give rise to marked improvements in CO₂RR activity and selectivity, relative to the bicarbonate/carbonate electrolytes.¹³

13. Li, Y. et al. Perovskite-socketed sub-3 nm copper for enhanced CO₂ electroreduction to C₂+. *Adv. Mater.* **34**, 2206002 (2022).”

3. “As flooding issue is inevitable in flow cell, the authors should account for the influence of flooding on catalyst performance and how to mitigate it.”

Response: We understand the concern of this reviewer. In general, the carbon-based gas diffusion layers (GDLs) are prevalently reported in the literature (*Chem. Soc. Rev.* **2020**, *49*, 7488), with hydrophobicity imposed within a carbon matrix *via* PTFE coating. As the reviewer pointed out correctly, the flooding issue of these GDLs is inevitable in flow cell, which typically happens within several hours of operation (*ACS Energy Lett.* **2021**, *6*, 33). If the GDL flooding occurs, CO₂ diffusion toward the catalyst layer is blocked and high CO₂RR rates can no longer be obtained (*J. Am. Chem. Soc.* **2021**, *143*, 8011). This is no exception in the present work. To be specific, our carbon-based GDLs suffered flooding issues when CO₂RR stability test of Sr₂CuWO₆ ran for more than 20000 s. Usually, for these GDLs, their flooding issue can be mitigated by washing away the carbonate precipitation. This has been widely used by many researchers to reactivate their electrodes (*ACS Energy Lett.* **2018**, *3*, 193). We have added corresponding notes to illustrate the flooding issue in the revised manuscript (page 12), as shown below.

Page 12, line 300-306:

“As a note, our gas diffusion layer suffered flooding issues when the CO₂RR stability test of Sr₂CuWO₆ catalyst ran for more than 20000 s. Since the flooding issues can cause an essential failure of the CO₂RR system, we terminated the electrolysis on the Sr₂CuWO₆ catalyst at about 20000 s; but the steady CO₂RR testing time for the Sr₂CuWO₆ catalyst itself was supposed to be for more than 20000 s by considering the well-stabilized Cu sites. Usually, the flooding issues

can be mitigated by washing away the carbonate precipitation. This has been widely used to reactivate the electrodes.^{42,43}

42. Verma, S. et al. Insights into the low overpotential electroreduction of CO₂ to CO on a supported gold catalyst in an alkaline flow electrolyzer. *ACS Energy Lett.* **3**, 193-198 (2018).
43. Rabiee, H. et al. Gas diffusion electrodes (GDEs) for electrochemical reduction of carbon dioxide, carbon monoxide, and dinitrogen to value-added products: a review. *Energy Environ. Sci.* **14**, 1959-2008 (2021).”

4. “How to derive the band centers of Cu and O? Please add relevant information.”

Response: We thank the reviewer for raising an interesting point. In the present work, the band center of Cu 3d or O 2p was calculated using the following equation (*Angew. Chem. Int. Ed.* **2022**, *61*, e202212329):

$$E_t = \frac{\int_{-\infty}^{\infty} E \cdot T(E) dE}{\int_{-\infty}^{\infty} T(E) dE}$$

where T(E) is the density of state (DOS) of orbitals. E corresponds to the occupied state ranges below the fermi energy level (E_F) in DOS. We have added the details about band center calculation to the revised supplementary information (page S2-S3), as shown below.

Page S2-S3, line 52-56 (Supplementary Information):

“The band center of Cu 3d or O 2p was calculated using the following equation³:

$$E_t = \frac{\int_{-\infty}^{\infty} E \cdot T(E) dE}{\int_{-\infty}^{\infty} T(E) dE}$$

where T(E) is the density of states (DOS) of orbitals. E corresponds the occupied state ranges below the fermi energy level (E_F) in DOS.

3. Yi, J.-D., Gao, X., Zhou, H., Chen, W., Wu, Y. Design of Co-Cu Diatomic Site Catalysts for High-efficiency Synergistic CO₂ Electroreduction at Industrial-level Current Density. *Angew. Chem. Int. Ed.* **61**, e202212329 (2022).”

5. “The intermediates adsorption models of TS state and *COCO_H step in Fig 3a may be wrong, which should be exchanged.”

Response: We thank the reviewer for the constructive suggestion. We have corrected the adsorption models of TS state and *COCO_H in the revised manuscript (Fig. 3a and Supplementary Fig. 8), as shown below. Moreover, to obtain a more rigorous analysis, we have provided a comprehensive assessment of the energetic profiles of reaction paths for both CH₄ and C₂H₄ production starting from *CO in the revised manuscript. Please refer to our response to the 5th comment from Reviewer #2.

Page 10 and Page S13:

“Fig. 3. (a) DFT-calculated energy diagrams for CH₄ and C₂H₄ formation on Sr₂CuWO₆(001) surface starting with *CO.

Supplementary Fig. 8. DFT-calculated structural details of intermediates on the Sr₂CuWO₆(001) surface.”

Reviewer #2

“In the manuscript entitled “ Superexchange-Stabilized Long-Distance Cu Sites in Rock-Salt-Ordered Double Perovskite Oxides for CO₂ Electromethanation,” the authors assert that the superexchange interaction contributes to enhancing the stability of oxide perovskite electrocatalysts. They propose that the superexchange interaction facilitates electron transfer from Cu to W, thereby reducing the electron density at the Cu site and modulating catalytic activity and selectivity. To support these claims, the authors present XPS and XANES spectra in conjunction

with the results of density functional theory (DFT) calculations.

However, I am hesitant to conclude that there exists clear experimental or computational evidence establishing the superexchange interaction as a pivotal factor during catalysis. The only aspect that can be unequivocally supported by experimental and theoretical data is that the Cu site in Sr₂CuWO₆ exhibits lower electron density than CuO. Nevertheless, there is no direct evidence demonstrating whether this reduction is attributable to the superexchange interaction or how it might influence changes in catalytic performance. It is imperative that we distinguish between correlation and causation. Consequently, I find the present manuscript to be premature, and I do not support its publication. My detailed comments are provided below: ”

Response: We greatly appreciate the reviewer’s efforts on reviewing our manuscript and providing us with constructive comments and suggestions to further improve the quality. Probably due to some imperfections in the expression of our manuscript, the reviewer made some misunderstandings. These misunderstandings include “*They propose that the superexchange interaction facilitates electron transfer from Cu to W, thereby reducing the electron density at the Cu site and modulating catalytic activity and selectivity*” and “*there is no direct evidence demonstrating whether this reduction is attributable to the superexchange interaction or how it might influence changes in catalytic performance*”. In the present work, we did not link CO₂RR activity/selectivity with superexchange interaction, but rather indicated the activity/selectivity for CH₄ closely related to the long Cu-Cu distance in Sr₂CuWO₆ (please see page 8-10, the subsection of Activity and selectivity for CH₄). And we also did not attribute the lowered electron density of Cu site into the superexchange interaction because of no causal relation between them (please see page 6-8, the subsection of Superexchange Interaction). But we just demonstrated the Cu and W sites bridged through the intermediate O sites and the existence of O-anion-mediated charge redistribution between Cu and W cations (*i.e.*, Cu-O-W superexchange interaction) in Sr₂CuWO₆, by using experimental characterizations (e.g., XRD, TEM, XPS, and XAS) and DFT calculations. This O-anion-mediated charge interaction has also been well demonstrated by many other studies (*Nat. Commun.* **2020**, *11*, 5657; *Phys. Rev. Lett.* **2020**, *124*, 077202; *J. Am. Chem. Soc.* **2018**, *140*, 11165; *Small* **2019**, *15*, 1903120; *Nat. Commun.* **2018**, *9*, 1085). We thank the reviewer for the constructive comments and suggestions again. All the suggestions and comments have been carefully addressed and revisions have been made accordingly.

1. “(Page 3, lines 55-56) *The statement that Cu-based perovskite oxides could be active towards CO₂ electromethanation by partially or entirely occupying the B sites is highly speculative. This argument lacks a solid logical foundation and lacks proper citations to substantiate this claim.*”

Response: We thank the reviewer for raising a good point. Up to date, in all categories of popular catalysts, Cu-based catalysts stand out, due mainly to the distinctive property of their Cu sites to generate appreciable CH₄ *via* stabilizing and subsequently hydrogenating the *CO species during CO₂RR (*Chem. Rev.* **2019**, *119*, 7610). For the perovskite oxides, upon most occasions, the nature of their B-site cations or B-O bonding determines their electrocatalytic properties in essence (*Science* **2017**, *358*, 751; *Adv. Funct. Mater.* **2021**, *31*, 2101872; *Acc. Mater. Res.* **2021**, *2*, 477). According to these characteristics of Cu-based catalysts and perovskite oxides, it can be reasonably inferred that if the B sites could be occupied entirely or partly by Cu element, the corresponding Cu-based perovskite oxides would be active toward CO₂ electromethanation. This inference has

been evidenced by typical examples involving Cu-based Ruddlesden-Popper perovskite oxides (e.g., $\text{La}_2\text{CuO}_{4-\delta}$) (*Angew. Chem. Int. Ed.* **2022**, *61*, e202111670; *ACS Catal.* **2020**, *10*, 4640; *Nano Lett.* **2021**, *21*, 980). Therefore, we believe that the statement that “*Inspired by this, if the B sites could be occupied entirely or partly by Cu element, the corresponding Cu-based perovskite oxides would be active toward CO_2 electromethanation*” is well-founded in the present work. To avoid misunderstandings, we have revised this statement and added the related references into the revised manuscript (page 3), as shown below.

Page 3, line 54-56:

“Based on the above characteristics of Cu-based catalysts and perovskite oxides, if the B sites could be occupied entirely or partly by Cu element, the corresponding Cu-based perovskite oxides would be active toward CO_2 electromethanation.”¹⁷⁻²²

17. Zhu, J. et al. Cation-deficiency-dependent CO_2 electroreduction over copper-based ruddlesden-popper perovskite oxides. *Angew. Chem. Int. Ed.* **61**, e202111670 (2022).
18. Wang, J. et al. Grain-boundary-engineered La_2CuO_4 perovskite nanobamboos for efficient CO_2 reduction reaction. *Nano Lett.* **21**, 980-987 (2021).
19. Schwartz, M. et al. Carbon-dioxide reduction to alcohols using perovskite-type electrocatalysts. *J. Electrochem. Soc.* **140**, 614-618 (1993).
20. Chen, S. et al. Highly selective carbon dioxide electroreduction on structure-evolved copper perovskite oxide toward methane production. *ACS Catal.* **10**, 4640-4646 (2020).
21. Singh, R. P. et al. Electrochemical insights into layered La_2CuO_4 perovskite: Active ionic copper for selective CO_2 electroreduction at low overpotential. *Electrochimica Acta* **326**, 134952 (2019).
22. Mignard, D. et al. Revisiting strontium-doped lanthanum cuprate perovskite for the electrochemical reduction of CO_2 . *Journal of CO_2 Util.* **5**, 53-59 (2014).”

2. “(page 3, line 71-73) “*The formation of double perovskite ... in CH_4 production.*”: This claim lacks a clear basis and appears overly speculative.”

Response: We would like to thank the reviewer for raising an interesting point but respectfully disagree with his/her assessment. In the present work, just behind the statement “*The formation of double perovskite structure ... affecting activity, selectivity, and stability in CH_4 production*”, we have given detailed and reasonable explanations for it. These explanations, which can be seen in the present work (page 4), are as follows: “*Specifically, in the rock-salt-type arrangement, the B-site cations alternate in all three crystallographic dimensions, markedly widening the distance between adjacent Cu cations, theoretically almost doubling relative to the undoped ones.*²⁸⁻³⁰ This increased distance could suppress $\ast\text{CO}$ dimerization and promote activity and/or selectivity for CH_4 production.^{31,32} Moreover, the B-site rock-salt ordering could bring superexchange interaction between Cu and B’ cations (mediated by intermediate O anions), giving rise to the redistribution of charge densities of the B-site cations via electron transfer.^{33,34} These superexchange interaction may hinder electron accumulation around the Cu sites and stabilize the Cu sites, thereby boosting the catalytic stability during CO_2RR ”. Therefore, we believe that our statement is well-founded rather than lacking a clear basis. We have added some related citations into the revised manuscript, as shown below.

Refs 28-34:

- “28. Yin, W.-J. et al. Oxide perovskites, double perovskites and derivatives for electrocatalysis, photocatalysis, and photovoltaics. *Energy Environ. Sci.* **12**, 442-462 (2019).
29. Xu, X., Zhong, Y. & Shao, Z. Double perovskites in catalysis, electrocatalysis, and photo(electro)catalysis. *Trends Chem.* **1**, 410-424 (2019).
30. Anderson, M. T., Greenwood, K. B., Taylor, G. A. & Poeppelmeier, K. R. B-cation arrangements in double perovskites. *Prog. Solid State Chem.* **22**, 197-233 (1993).
31. Peng, C. et al. Highly-exposed single-interlayered Cu edges enable high-rate CO₂-to-CH₄ electrosynthesis. *Adv. Energy Mater.* **12**, 2200195 (2022).
32. Guan, A. et al. Boosting CO₂ electroreduction to CH₄ via tuning neighboring single-copper sites. *ACS Energy Lett.* **5**, 1044-1053 (2020).
33. Katukuri, V. M. et al. Exchange interactions mediated by nonmagnetic cations in double perovskites. *Phys. Rev. Lett.* **124**, 077202 (2020).
34. Tong, Y. et al. Vibronic superexchange in double perovskite electrocatalyst for efficient electrocatalytic oxygen evolution. *J. Am. Chem. Soc.* **140**, 11165-11169 (2018).”

3. “(page 4, line 80-81) “These superexchange interaction may hinder electron accumulation around the Cu sites, ... , thereby boosting the catalytic activity during CO₂RR.”: To the best of my knowledge, the superexchange interaction primarily plays a role in determining the spin arrangements within the system and is not strongly associated with the degree of electron accumulation. Furthermore, there is no established mechanistic link between the superexchange interaction and catalytic activity.”

Response: We thank the reviewer for raising interesting points. It is our impression that the reviewer may misunderstand the sentence in our original manuscript. In our original manuscript, the sentence is “These superexchange interaction may ..., thereby boosting the **catalytic stability** during CO₂RR” rather than “These superexchange interaction may ..., thereby boosting the **catalytic activity** during CO₂RR”. In the present work, our original intention was not to claim that the superexchange interaction was strongly associated with the degree of electron accumulation, but to illustrate that during CO₂RR, the superexchange interaction may transfer the electrode-supplied electrons accumulated around the Cu sites to B’ sites and stabilize the Cu sites, thereby boosting the catalytic stability. To avoid misunderstandings, we have rewritten the relevant statement in the revised manuscript (page 4), as shown below.

Page 4, line 80-82:

“During CO₂RR, this superexchange interaction may transfer the electrode-supplied electrons accumulated around the Cu sites to B’ sites and stabilize the Cu sites, thereby boosting the catalytic stability.”

Furthermore, in the present work, we do not link CO₂RR activity/selectivity with superexchange interaction, but rather indicate that the activity/selectivity for CH₄ is closely related to the long Cu-Cu distance in Sr₂CuWO₆. As a note, in some other electrocatalysis fields (e.g., oxygen reduction, hydrogen evolution, oxygen evolution), the mechanistic link between the superexchange interaction and catalytic activity has been well established by previous studies (*Adv. Mater.* **2018**, *30*, 1705407; *Nat. Commun.* **2020**, *11*, 5657; *Small* **2019**, *15*, 1903120; *J. Am. Chem. Soc.* **2018**, *140*, 11165). In future research, we will also try to establish the mechanistic link between CO₂ property and superexchange interaction in double perovskite oxides.

4. “(page 7, line 168) The authors argued that the distinct partial charge distributions at the Cu and W sites of Sr₂CuWO₆, as compared to CuO+WO₃, are mediated by O sites. In order to substantiate the existence of a superexchange interaction, it is crucial to demonstrate the involvement of the 2p orbitals of the O sites. However, there is currently no compelling evidence presented to establish that O atoms indeed mediate the charge redistribution as claimed.”

Response: We would like to thank the reviewer for raising an interesting point but respectfully disagree with his/her assessment. In the present work, our XPS and XAS spectra fully demonstrated the existence of charge transfer between Cu²⁺ and W⁶⁺ sites (Fig. 2a-f). Such charge transfer was most likely attributed to the electronegativity difference between Cu²⁺ (1.372) and W⁶⁺ (2.175) (*J. Phys. Chem. A* **2006**, *110*, 11332). According to crystal structure characterizations (Fig. 1a-e), for the Sr₂CuWO₆, its Cu and W sites were bridged through the intermediate O sites in all three crystallographic dimensions rather than directly connected with each other. Combined with the above analyses, the charge transfer between rock-salt-ordered CuO₆ and WO₆ octahedra must be mediated by the intermediate oxygen anions, thus being defined as superexchange interaction (*Adv. Mater.* **2018**, *30*, 1705407). As a note, in (double) perovskite oxides, such O²⁻-mediated electron interaction between B and B' cations (*i.e.*, superexchange interaction) has been well investigated by many previous studies (*Nat. Commun.* **2020**, *11*, 5657; *Phys. Rev. Lett.* **2020**, *124*, 077202; *J. Am. Chem. Soc.* **2018**, *140*, 11165; *Small* **2019**, *15*, 1903120; *Nat. Commun.* **2018**, *9*, 1085). Moreover, our theoretical calculations further verified the charge redistribution between Cu and W sites mediated by O sites in Sr₂CuWO₆ (Fig. 2g and Supplementary Table 3). To be specific, in Fig. 2g, the light-blue regions, surrounding the Cu, O, and W sites, clearly depicted the Cu-O-W charge transfer channels. Taken together, we believe that our work has given sufficient evidence and reasonable analysis to demonstrate the existence of O²⁻-mediated charge redistribution in Sr₂CuWO₆. To avoid misunderstandings, we have rewritten the relevant statement in the revised manuscript (page 7) and also have added the related discussion on Cu-O-W charge transfer channels (Fig. 2g) into the revised manuscript (page 7), as shown below.

Page 7, line 168-170:

“Combined with the above crystal structure characterization, one can believe that this electron transfer between rock-salt-ordered CuO₆ and WO₆ octahedra must be mediated by the intermediate oxygen anions, thus being defined as superexchange interaction.”

Page 7, line 172-174:

“In Fig. 2g, the light-blue regions, surrounding the Cu, O, and W sites, clearly depicted the Cu-O-W charge transfer channels.

Fig. 2. (g) Top view of charge distribution between CuO_6 and WO_6 octahedra in Sr_2CuWO_6 . Cu, W, and O are represented by blue, gray, and red dots, respectively. The light-blue regions (surrounding the Cu, O, and W sites) depict the electron transfer channels.”

5. “(Figure 3a) In Figure 3a, the authors juxtapose the reaction energy difference for the CH_4 production path with the barrier energy for the ethylene production path in order to draw a comparison. However, this approach represents a biased comparison and does not provide a comprehensive assessment of the energetics for the entire reaction pathways. A more rigorous analysis would involve comparing the energetic profiles of the full reaction paths for both CH_4 and ethylene production.”

Response: We thank the reviewer for the very constructive suggestion. In the revised manuscript, we have re-performed DFT calculations to predict CO_2RR properties of Sr_2CuWO_6 catalyst (Fig. 3a and Supplementary Fig. 8). In the new DFT calculations, the full reaction pathways for CH_4 and C_2H_4 formation starting from CO^* were constructed, which were in line with the representative previous studies (*Nat. Catal.* **2022**, 5, 564; *Nat. Commun.* **2022**, 13, 1877). On the $\text{Sr}_2\text{CuWO}_6(001)$ surface, the rate-determining step for CO^* to CH_4 is CO^* hydrogenation to CHO^* , and that for CO^* to C_2H_4 is $\text{CO}^*\text{-COH}^*$ dimerization (Fig. 3a). The energy barrier for the CO^* hydrogenation was about 0.64 eV, lower than that (0.77 eV) of the $\text{CO}^*\text{-COH}^*$ dimerization. This also reveals a preference for CH_4 formation. We have added new discussions on DFT calculations into the revised manuscript (page 9), as shown below.

Page 9, line 207-213:

“We took the full reaction pathways for CH_4 and C_2H_4 formation starting from CO^* as analysis objects and calculated their corresponding energy profiles at the Cu sites.^{39,40} On the $\text{Sr}_2\text{CuWO}_6(001)$ surface, the rate-determining step for CO^* to CH_4 is CO^* hydrogenation to CHO^* , and that for CO^* to C_2H_4 is $\text{CO}^*\text{-COH}^*$ dimerization (Fig. 3a and Supplementary Fig. 8). The energy barrier for the CO^* hydrogenation was about 0.64 eV, lower than that (0.77 eV) of the $\text{CO}^*\text{-COH}^*$ dimerization, revealing a preference for CH_4 formation.

39. Xie, Y. et al. High carbon utilization in CO_2 reduction to multi-carbon products in acidic media. *Nat. Catal.* **5**, 564-570 (2022).

40. Liu, W. et al. Electrochemical CO_2 reduction to ethylene by ultrathin CuO nanoplate arrays. *Nat. Commun.* **13**, 1877 (2022).

Fig. 3. (a) DFT-calculated energy diagrams for CH₄ and C₂H₄ formation on Sr₂CuWO₆(001) surface starting with *CO.

Supplementary Fig. 8. DFT-calculated structural details of intermediates on the Sr₂CuWO₆(001) surface.”

Reviewer #3

“Jiawei Zhu & Heqing Jiang *et al* present a Cu-based double perovskite oxides A₂CuB’O₆ with B-site rock-salt ordering to boost CO₂-to-CH₄ conversion, inhibiting C-C coupling reactions. As model catalyst, the authors synthesised Sr₂CuWO₆, exhibiting corner-linked CuO₆ with high Cu-Cu distances. Indeed, the performance of the systems for CO₂RR is remarkable with a ~73% FE and -400mA/cm², which is a competitive number, but does not stand out in the field.”

Response: We thank the reviewer for the great comment. As the reviewer pointed out correctly, the Sr₂CuWO₆ was the most effective Cu-based-perovskite catalyst for CO₂-to-CH₄ conversion. And in the whole CO₂RR field, although the activity and selectivity for CH₄ of Sr₂CuWO₆ were not the best, they were very close to the current highest level (FE_{CH₄} = 81%) (*Angew. Chem. Int. Ed.* **2022**, *61*, e202203569). In future research, we will further design novel Cu-based double perovskite oxides based on the present work to achieve breakthroughs in CO₂RR performance.

“The reviewer noted the remarkably long-term stability of the Sr₂CuWO₆ at reductive potentials. The authors proposed that high stability stems from a super exchange stabilisation of the Cu sites, proposed via O-anion-mediated ET from Cu to W6 cations. This reviewer is not found of this explanation as other factors were not considered. However, the presented calculations and rationale do support the claims of the authors. In situ and ex situ spectroscopic experiments support that the material does not undergo dramatic transformation, which is unique.”

Response: We thank the reviewer for his/her high recognition of the long-term stability as well as the related mechanism analysis in our work.

“It should be noted that this reviewer cannot in detail comment on the DFT calculations performed. This reviewer noted that methods and approaches (Bader charge calculations etc.) employed presented in this manuscript are in line with other works in this field.”

Response: We thank the reviewer for the great comment. As the reviewer commented, in the present work, the methods and approaches of DFT calculations, in line with previous works, are reasonable and well-founded. Notably, to obtain a more rigorous analysis, we have provided a comprehensive assessment of the energetic profiles of reaction paths for both CH₄ and C₂H₄ production starting from *CO in the revised manuscript. For the details, please refer to our response to the 5th comment from Reviewer #2.

“The manuscript is clearly outlined and very well written. Synthesis of the Sr₂CuWO₆ is well described. The structural characterisation has been carried out with care and seems to verify the structure of the proposed Sr₂CuWO₆ the structure. The spectroscopic experiments are well presented.”

Response: The reviewer positively recognized our work: “...is clearly outlined and very well written. ... carried out with care. ...experiments are well presented.” We thank the reviewer for the great comments.

“This reviewer believes that the work and the basics of the research is original, and the performance, i.e., high FE and j(CH₄), long-term stability, is competitive. Thus, the work will be of interest for those working in the field of electrocatalysis, perovskites, and CO₂RR, and beyond.”

Response: The reviewer highly praised our manuscript as follows: “the work and the basics of the research is original, and the performance...is competitive. ...the work will be of interest...”. We thank the reviewer for the great comments.

REVIEWER COMMENTS

Reviewer #1 (Remarks to the Author):

All my concerns have been addressed in the revised ms.

Reviewer #2 (Remarks to the Author):

The authors addressed the majority of my earlier inquiries and clarified some of my misunderstandings. Also the full reaction free energy profiles are informative. However, I have one remaining concern. The authors referred to both energy quantities related to the $^*CO-^*CHO$ step and the $^*CO+^*COH-TS$ step as "energy barriers." However, this is inaccurate; the former represents an energy difference between intermediate states, while the latter is specifically a barrier quantity.

Regarding this matter, the energy required for the *CO hydrogenation, reported as 0.64 eV, serves as the minimum threshold for the barrier in methane production. On the other hand, I think that it is more reasonable to consider the energy barrier for ethylene production as the difference from the locally most stable state, i.e., 2^*CO , to the highest TS, resulting in a total of $0.31+0.77 = 1.08$ eV. Therefore, the methane production still remains a favorable option, presuming that the actual energy of transition state for the *CO hydrogenation is not significantly different from the energy of the *CO step.

While the final conclusion remains unchanged, I believe it is important to explicitly articulate this distinction.

Response Letter to Reviewers

Reviewer #1

“All my concerns have been addressed in the revised ms.”

Response: The 1st reviewer considered our revised manuscript has addressed the reviewers' concerns. We thank the reviewer for the great comment.

Reviewer #2

*“The authors addressed the majority of my earlier inquiries and clarified some of my misunderstandings. Also the full reaction free energy profiles are informative. However, I have one remaining concern. The authors referred to both energy quantities related to the *CO-*CHO step and the *CO+*COH-TS step as "energy barriers." However, this is inaccurate; the former represents an energy difference between intermediate states, while the latter is specifically a barrier quantity.*

*Regarding this matter, the energy required for the *CO hydrogenation, reported as 0.64 eV, serves as the minimum threshold for the barrier in methane production. On the other hand, I think that it is more reasonable to consider the energy barrier for ethylene production as the difference from the locally most stable state, i.e., 2*CO, to the highest TS, resulting in a total of $0.31+0.77 = 1.08$ eV. Therefore, the methane production still remains a favorable option, presuming that the actual energy of transition state for the *CO hydrogenation is not significantly different from the energy of the *CO step.*

While the final conclusion remains unchanged, I believe it is important to explicitly articulate this distinction.”

Response: We greatly thank the reviewer for his/her constructive comment and suggestion. According to the reviewer's suggestion, we have revised the related statements in the revised manuscript (page 3), as shown below.

Page 9, line 209-214:

*“On the Sr₂CuWO₆(001) surface, the energy difference between *CO and *CHO was about 0.64 eV, much lower than the energy barrier (1.08 eV) for C₂H₄ production (i.e., 2*CO to the TS) (Fig. 3a and Supplementary Fig. 8). As a result, CH₄ formation was more favorable on the Sr₂CuWO₆(001) surface based on the presumption that the energy of TS for the *CO hydrogenation was not significantly different from the energy of *CO step.”*